# Bowel Sounds Identification and Migrating Motor Complex Detection with Low-Cost Piezoelectric Acoustic Sensing Device

**DOI:** 10.3390/s18124240

**Published:** 2018-12-03

**Authors:** Xuhao Du, Gary Allwood, Katherine Mary Webberley, Adam Osseiran, Barry J. Marshall

**Affiliations:** 1The Marshall Centre for Infectious Diseases Research and Training (M504), The University of Western Australia, Crawley, WA 6009, Australia; gary.allwood@uwa.edu.au (G.A.); mary.webberley@uwa.edu.au (K.M.W.); admin@hpylori.com.au (B.J.M.); 2School of Engineering, Edith Cowan University, Joondalup, WA 6027, Australia; a.osseiran@ecu.edu.au

**Keywords:** bowel sound, migrating motor complex, piezoelectric, quantity proportion

## Abstract

Interpretation of bowel sounds (BS) provides a convenient and non-invasive technique to aid in the diagnosis of gastrointestinal (GI) conditions. However, the approach’s potential is limited by variation between BS and their irregular occurrence. A short, manual auscultation is sufficient to aid in diagnosis of only a few conditions. A longer recording has the potential to unlock additional understanding of GI physiology and clinical utility. In this paper, a low-cost and straightforward piezoelectric acoustic sensing device was designed and used for long BS recordings. The migrating motor complex (MMC) cycle was detected using this device and the sound index as the biomarker for MMC phases. This cycle of recurring motility is typically measured using expensive and invasive equipment. We also used our recordings to develop an improved categorization system for BS. Five different types of BS were extracted: the single burst, multiple bursts, continuous random sound, harmonic sound, and their combination. Their acoustic characteristics and distribution are described. The quantities of different BS during two-hour recordings varied considerably from person to person, while the proportions of different types were consistent. The sensing devices provide a useful tool for MMC detection and study of GI physiology and function.

## 1. Introduction

For centuries, auscultation has been used as a non-invasive technique in medicine [1]. The most commonly studied organs were, and still are, the lungs and the heart, with doctors long using sounds to diagnose disease in the cardiovascular and respiratory systems [2,3,4]. Sounds have more recently been exploited as biomarkers in fields as diverse as joint ageing and degeneration [5], and placental pathophysiology [6].

Doctors have also listened to gut noises to diagnose conditions such as bowel obstructions and paralytic ileus based on the nature or absence of bowel sounds (BS) [4]. In 1905, the first scientific study of bowel auscultation was reported by Cannon [7]. Cannon found that the gut produced rhythmic noises, most likely from the peristaltic movement of the intestines, plus continuous random noises varying in location and intensity. His study prompted many researchers to try to understand the relationship between BS and gastrointestinal (GI) physiology and disease. To date, automatic or computerized gut-noise-based diagnosis and prognosis has not been widely developed or adopted. However, detected BS are known to have distinctive characteristics, indicating that they are generated by different bowel movements. Therefore, BS must contain meaningful information about the processes inside the bowel which could be analyzed systematically [8,9]. The irregular pattern and occurrence of BS means that the potential of short recordings is limited and that methods allowing longer recordings of BS are required to fully exploit this approach.

BS generation is linked with gastrointestinal motility and a key physiological phenomenon underlying motility, the migrating motor complex (MMC) [10]. The MMC is a cyclic, recurring motility pattern that occurs in the stomach and small bowel during fasting. It has a housekeeping role, ensuring contents move forward through the gastrointestinal tract. The MMC repeats every 80 to 150 min [10] with three different phases. The detection of MMC cycle can have significant clinical importance [11,12]. During the fasting state, the absence of MMC might indicate small intestinal bacterial overgrowth, due to a motility disorder. It is also commonly measured during clinical trials of new drugs developed to alter motility [10]. Although monitoring of the MMC is important, detection of the MMC and other measure of motility involves expensive, invasive, and uncomfortable procedures [12,13]. For example, the MMC is usually detected using antropyloroduodenal manometry. The method typically uses water-perfused manometric catheters or solid-state sensors mounted on motility catheters. Tube placement usually requires upper GI endoscopy and skilled technical support [12].

The relationship between MMC and BS was first proposed by Tomomasa et al. in 1999 [10]. They found that the sound index, which is the sum of absolute signal amplitudes, links with the MMC cycle due to the BS generation mechanism. Within a five-hour recording, they observed two MMC cycles. Thus, this sound index shows potential as a biomarker for clinical use. The BS can be divided into small popping sounds and huge gurgling sounds, which are significantly different in amplitude. Therefore, we hypothesize that the sound index might be dominated by the gurgling sounds due to their large amplitude. Also, it is important to identify different kinds of BS in the different phases of MMC to better understand if the sound index could act as a reliable biomarker for the MMC. This paper proposes to use the sound duration for indicating either the popping sound or large sound and identify the dominant sound during MMC phases. We also demonstrate the effectiveness of our simple and low-cost acoustic sensing device designed for long recordings of BS.

Microphone-based sensors are popular for BS recording [14,15,16,17,18,19,20]. In 2013, Sakata et al. developed a silicon microphone-based sensor for long-term BS recording [15]. They studied sound occurrence over time before and after eating. Kim et al. used a similar microphone-based sensor attached to the abdomen for long-term BS recordings [16,21]. They extracted the jitter and shimmer from BS for gut motility estimation. Spiegel et al. built a microelectronic microphone for gut motility monitoring via BS [17]. Their sensor and associated algorithm can distinguish healthy controls from patients recovering from abdominal surgery and predict which patients will develop postoperative ileus. Use of a microphone for BS recording is straightforward. However, it requires a power supply, which could cause inconvenience. Since microphone-based sensors are sensitive to airborne noise, the ambient noise might easily contaminate the BS signal. Emoto et al. proposed a non-contact microphone for BS recorded and used their system to identify gut motility before and after soda intake [19,20]. However, their participants were required to lie down on a bed for recordings, which is inconvenient for long recordings. A piezoelectric-based sensor is another choice for passively recording physiology focused sound and vibration [22,23]. Dimoulas et al. have done much research using long recording of BS signals, which were recorded by piezoelectric-based sensor [24,25]. However, the exact structure of their sensor is never mentioned, and their studies mainly focused on signal processing and BS types classification. Due to the way BS are conducted through the abdomen, a piezoelectric-based sensing device is suitable for BS recording as it is more sensitive to vibration trigger signals and can sit on the skin to pick these up. Furthermore, it is much less sensitive to ambient air borne noise. In this study, a piezoelectric acoustic sensing device was developed and decomposed for demonstration. Its design makes it low cost, easy to make, less noise contaminated, and accurate enough for longer BS recordings, making it suitable for research and probably clinical use.

The proposed device can be evaluated by comparing the observed BS with previous studies. Different BS classification methods have been investigated, although no standard definitions or classes of BS have been established. In 1967, Watson and Knox described three different types of BS by analyzing the sounds from three participants [26], including BS with a regular pattern, rising pitch and crescendo intensity, BS with unregulated patterns, and tinkling BS. Dalle et al. also reported three types of BS, using their durations as the basis of a classification index [27]. Dalle et al. are also noteworthy because they pioneered the use of computers to analyze BS. Dimoulas et al. classified BS into five different categories based on their waveforms and perceived characteristics [25]. These five types comprised solitary clicks, repeated clicks, sequences of irregularly concatenated segments, crepitating sweeps, and whistling sweeps. The authors also linked them to different gut activities. Ulusar recognized BS as single bursts and multiple bursts based on their waveform to build a BS classification model using Bayesian theory. Du et al. also proposed a mathematical model for BS generation to explain different types of BS [28]. As summarized above, there are several classifications of BS based on different sound characteristics, and the quantities of different types of BS were not studied in their work. However, the quantities may be important for gut motility detection. The most common classification indexes include duration, frequency, waveform, auditory perception, and mechanisms for the production of BS. This paper will also summarize different types of BS based on our observations. We also analyzed the quantities of the BS types across and within participants.

This study presents a low-cost and straightforward piezoelectric acoustic sensing device that can effectively allow detection of BS and MMC cycle with a personal computer and commercial software Audacity. This offers the potential of greater understanding of the physiology of the gastrointestinal tract and possible clinical use. BS from ten participants were recorded for two hours for identification and characterization of BS types. The observed BS were classified in a fundamental way with two physical indexes, waveform, and spectrogram. Five different types of BS were systematically analyzed including the duration, spectral flatness, spectral bandwidth, and mean-crossing ratio. The quantities of different types of BS were also counted from these long-term recordings, and their quantities proportion of different types within each participant were found to be stable. Also, in eight- and four-hour recordings from a healthy participant, several MMC cycles and different phases were observed under the fasting state using the sound index, and longer BS are more likely to occur in Phase III using sound duration as another index. As expected, the cycles disappeared when the participant consumed food.

## 2. Material and Method

### 2.1. Sensor Design

A low-cost and straightforward piezoelectric sensing head that connects to a sound recorder (e.g., a personal computer) was used for BS recordings. This piezoelectric sensing head consisted of a solid housing, a membrane, a piece of foam and a piezoelectric disk. The piezoelectric disk is composed of a PZT ceramic circle on the top of brass circular base, with diameters of 20 mm and 27 mm, respectively. The thickness of the disk is 0.52 mm and the resonant frequency and impedance are 4200 Hz and 300 Ω, respectively. This piezoelectric disk is commercially available, and the reader could buy it online. The sensor head configuration is shown in Figure 1 with dimensions. Due to the stimuli mechanism of piezoelectric sensors, which are triggered by vibration, the housing can effectively block out the ambient noise. In addition, the housing was made from aluminum material, which effectively suspends friction noise from the belt and clothes due to its smoothness. A piece of foam was inserted inside underneath the top lid to push down the piezoelectric sensor and membrane therefore establishing a better contact with the abdomen. The piezoelectric sensor was attached to a membrane at the center with a dot of glue. Subsequently, the BS related vibrations of the abdomen can be transferred to the piezoelectric disc effectively. The sensor was compared to that of a state-of-the-art electronic stethoscope (Litmann 3200) when listening to BS and showed a similar performance. A stretchy tubi-grip belt was used to hold the sensor head at the right location. This low-cost and straightforward piezoelectric sensing device can effectively detect the BS with high accuracy compared to sensors used in previous research. Multiple bio-indexes can be observed such as sound index, sound duration, and the MMC.

### 2.2. Preliminary Verification

Two experiments were conducted for BS verification. An anechoic chamber recording on the body was conducted to ensure that the types of sounds that we had identified as coming from the abdomen were not actually background noise from our quiet recording room. The sounds were present in both locations, which indicated that the sounds were indeed BS. Another preliminary recording in our quiet room was executed and a potential BS library was established for two medical doctors to verify. The BS in the library was collected from the participants’ recordings with 18 BS containing all different types and eight irrelevant and environmental noises. Two medical doctors participated in this blind test. 100% of the BS and noises identified in this study matched the doctor’s judgement based on their experience. Hence, we systematically validated the recorded BS as sounds typically regarded as GI in origin by clinicians.

### 2.3. Experiment Setup

Two experiments were designed to record BS passively for BS identification and MMC detection and observation, with the study approved by the UWA Human Research Ethic Office (study no. RA/4/1/8893).

First, two-hour recordings were taken from each of ten participants (participant No. 1 to 10) with 44.1 kHz sampling frequency. This quantity of participants was selected because it was found to be useful in previous studies of the individual characteristics of BS [26,27,29,30]. A single sensor head was attached to the lower quadrant of the abdomen. This site was selected to minimize interference from other organs (heart and lungs) and provide a large amount of BS (following Cannon [7]). To measure standardized ‘clean’ BS without any food influence, the participants were required to fast overnight and skip breakfast. They were asked to sit still in a quiet room for two hours during which sounds from the bowel were recorded. Subsequently, the 20 h of recordings were processed, and several features were extracted, analyzed, and compare to previous research findings.

Second, a total of twelve hours of BS recordings were taken from participant L, primarily for study of the MMC. Recordings were made using two sensor heads, one placed on the upper abdomen and one placed on the lower abdomen, simultaneously. Participant L was asked to fast overnight, skip breakfast and recorded under fasting conditions for eight hours to see the MMC cycle. Both the sound duration and sound index over the eight hours of recording were documented in this stage. Another four hours recording after a meal was conducted to investigate how the MMC changed after the meal.

## 3. Result and Discussion

### 3.1. Bowel Sound Categories

Using the 20 h of recordings from ten participants, we identified five typical types of BS with our proposed sensing device. They were identified according to their time and spectrogram information expanded based on short time Fourier analysis [31]. These five types of BS are classified as a single burst (SB), multiple bursts (MB), continuous random sound (CRS), harmonic sound (HS) and a combination sound (CS), as shown in Figure 2. We were subsequently able to link these BS types to other BS descriptions in the literature, which further validates the performance of our acoustic sensing device.

The most frequent type of BS in the recordings was the SB, which is a simple pulse probably caused by a single contraction of the bowel muscle [25,28,29]. An SB, with its distinctive peak frequencies, is noticeable in the time domain. An example of a SB is presented in Figure 2a with both the time and frequency domain representations.

In Figure 2a, the top figure represents the time domain signal and the lower figure represents its frequency spectrogram. The duration of the single burst is short, only 10–30 ms and no other SB is present within 100 ms on either side. There is usually a distinct peak frequency of the SB. The frequency of this example was around 400 Hz, but it can vary from 200 to 1000 Hz for different SB. The SB is comprehensively reported in the literature [27,29,32]. Since the SB occurs the most frequently, it therefore makes up the largest proportion of the total quantity of BS.

MB can be described as a repetitive SB with a shorter interval time between adjacent components. Figure 2b gives an example of this type of BS. Each component in the MB looks quite similar in the spectrogram with slight differences in bandwidth and amplitude, which indicates that the MB consists of several similar individual components. The quantities of the repetitive components are not consistent in MB, as they vary from two to dozens within a single MB. There are clear silent gaps between the adjacent components in the time domain and the length of these silent gaps are also inconsistent. The spectrogram of the MB is similar to the SB, although the duration of MB is much longer than the SB and ranges from 40 to 1500 ms. The MB is also comprehensively described in the literature [25,29].

The CRS is shown in Figure 2c. The waveform of the CRS is usually continuous over long periods of time ranging from 200 ms to 4000 ms without any defined silent gaps. The CRS is usually recognized as a random sound because it has no clear rhythm or pattern. The CRS waveform is also less regular compared to other types of BS although it occurs more often than the HS and CS. It is also clear that the CRS often appears in a combination of other types of BS to construct a CS. It appears that CRS are reported elsewhere in the literature references [7,25,33] with the names continuous random noises, crepitating sweeps, and prolonged sounds.

Another typical BS is the HS, which is a whistling-like sound and is presented in Figure 2d. Three to four clear frequency components appear in the spectrogram of the HS such as the harmonic sound, which causes the whistling-like sound. Those peak frequencies are multiples of the fundamental frequency, which is usually relatively low, around 200 Hz. The highest harmonics recorded in our experiment were up to 3000 Hz. The duration of the HS ranges from 50 ms to 1500 ms. In the time domain, a few peaks could be observed in the HS with no defined silent gaps between each peak. Sounds of the HS type infrequently occur (see below). HS have been described in other studies [7,25,26] with the descriptions: rhythmic noises, whistling sweeps, and regular pattern.

In addition to the above four separate types of BS, many detected BS often appear as a combination of the types described above. Figure 2e is from a BS where the first part is a CRS, while the end contains typical HS characteristics. The CRS and HS are not the only possible combination, every combination of the five BS described above are possible, and all were collected during the 20 h of recordings. These types of BS will usually last for a long time. They appear to have been previously described by Dimoulas et al. [25] with the name “irregularly concatenated segments”. Due to their inconsistent characteristics, their quantity and acoustic features were not directly counted and analyzed.

### 3.2. Quantity and Characteristics of Different Types of Bowel Sound

Table 1 presents the quantities and proportions of each type of BS recorded from the ten participants’ 2 h recording. An automation BS identification algorithm was used for BS counting. Because MB, CRS, HS, and CS rarely happened, they were counted manually from the identified BS. Therefore, the quantities of SB were calculated by subtracting the number of other types from the total numbers. The numbers in brackets represent the proportion of different types of BS.

As presented in Table 1, it is clear that the quantities of overall BS vary from person to person. Some participants had more than 5000 BS for two hours, corresponding to 0.7 BS every second, while some only had 300 for two hours, which is 0.04 per second. It has previously been reported that the quantity of BS varies across individuals [34]. BS are reduced and useful in the diagnosis of obstruction and in patients with prolonged colon transit time, such as patients with Parkinson’s disease and multiple system atrophy patients [35]. The quantities of BS are also reduced in patients with postoperative ileus after the abdominal surgery [18]. However, the detection of quantities reduction requires a baseline for comparison.

Although the quantity ratios across these four types of BS vary, their proportions were consistent. The consistency can be represented by CV, which is the ratio between the standard deviation and the mean value. It is clear that the CV of proportion is much smaller than the CV of quantities. Table 1 indicates that the most common type is SB, which makes up 87.8% of sounds on average (mean value) with only 5.4% SD. The second most abundant type is MB, which makes up on average 7.8% of sounds with 3.5% SD. The least type is HB, average only 0.9% and below 2% for all participants.

Several typical acoustic features were studied for different types of BS including the duration, spectral bandwidth, spectral flatness, and mean-crossing ratio. The BS acoustic features from first and third quadratic are shown in Table 2, and the distributions are shown in Figure 3. The amplitudes are normalized so that in all cases the area under the curve is equal to one.

For the duration, it is common that the SB is typically short and the duration of the other four types of BS have a similar distribution. However, the CRS has some extra-long BS beyond 4000 ms. The second longest type of BS is the MB, which can last up to 3000 ms. In this study, the longest duration observed for the HS was 1800 ms. The spectral centroid is the “center of mass” of the spectrum, where all five types of BS share similar distribution and values. The spectral bandwidth is defined as the wavelength interval in which a radiated spectral amplitude is not less than half its maximum value. The mean value of spectral bandwidth decreases from MB, SB, HS to CRS. The spectral bandwidth distribution of the CRS type of BS is the narrowest at low spectral bandwidth. The spectral flatness is a measure used in digital signal processing to characterize an audio spectrum to show how tonal the signal is. It is interesting to note that the CRS has the smallest spectral flatness. As with spectral bandwidth, the mean values of spectral flatness decrease from MB, SB, HS to CRS. The SB and HS share the same distribution with different mean values, and the CRS has the narrowest distribution again. This order is observed in the spectral bandwidth and mean-crossing ratio. The mean-crossing ratio value is the number of times the waveform crosses its mean value. The distributions of these five types of BS have the most significant difference among all other characteristics. The MB tend to have a higher value with narrower distribution. The lowest mean value is that of the HS with wide distribution.

### 3.3. Migrating Motor Complex Detection

The BS sound index was calculated in a long recording after verifying the performance of our acoustic sensing device and identifying the BS. The presented sound index values are scaled by the maximum values in Figure 4 and Figure 5 for clarification. Figure 4 and Figure 5a shows the scaled sound index over a long-term BS recording from participant L. BS were detected automatically during these eight- and four-hour recordings and the sound index calculated for each three minutes of recording. Also, each BS duration was measured and shown in Figure 4 as the red dashed line. From Table 2, it is clear that SB is significantly smaller than the other sounds so that the duration can be used effectively to separate the SB and others. To see the difference between different parts of the abdomen, the sensor was placed at both the lower and upper abdomen, shown in Figure 4a,b, respectively.

From Figure 4a,b, six cycles were detected in the eight hours of recording by assessing the sound index. The pattern was clearest in the recordings from the lower abdomen, which matches with Tomomasa’s observation [10]. As show in Figure 4a, the duration of six cycles varied from 67.4 min to 109.2 min with a mean of 79.5 ± 14 min, which is a normal value. Figure 4b presents the sound index at the lower abdomen. The cycles can also be observed in this region as well as its phase changes, from Phase I to Phase III, which are denoted by the grey brackets. During the MMC Phase I, which is the longest phase in MMC, little motor activity happened in the bowel. At Phase II, the gut activity starts to increase slowly. The gut activity then rapidly reaches its peak, which is denoted as Phase III. Studying BS duration in Figure 4a,b reveals that longer BS were generated in Phases II and III. Since longer BS likely indicate increased gut activity, this phenomenon appears to match well with the phases of activity known to occur during the MMC . Figure 5a presents the sound index and duration over four hours after a meal at the lower abdomen. During these four hours, the clear pattern disappears as the MMC is interrupted by food consumption, as expected. In addition, the sound index was much larger than under the fasting state. The long BS are also spread uniformly, which indicates high gut motility during this time. This is reasonable according to the mechanism of MMC [13].

In addition, the MMC pattern was also observed in the other ten participants from their two-hour recording based on the sound index. As shown in Figure 5b, the peaks of Phase III were presented from the sound index of the first three participants and the same observation of the sound duration was obtained. Also, two Phase III peaks with 77.2 min interval time were observed in Participant No. 2. The preliminary tests further demonstrate the utility of our simple, non-invasive acoustic sensing device.

We also studied the mean BS quantities before and after the meal over the eight- and four-hour recording. The means were 3.09 and 7.12 BS per minute for the upper and lower abdomens respectively, from the first eight hours of recording. From the recording after meal, there were on average 4.59 and 45.88 BS per minute detected at the upper and lower abdomen, respectively. These quantities indicate a significant increase in gut activity in the lower abdomen after eating.

## 4. Conclusions

A low-cost and straightforward acoustic sensing device was proposed in this paper to enable long duration BS recordings. The sensor head consisted of only a piezoelectric sensor, housing, membrane, and a piece of foam. The performance of the device has been tested with two hours of recording from each of the ten participants and an eight- and four-hour recording from one participant. We undertook analysis of BS from ten volunteers with the proposed sensing device. Our simple sensor heads were able to pick up previously described BS types. However, we were also able to improve characterization of BS types and produce a standard categorization system. Based on our analysis, we found five different characteristic types of BS based on their waveform and spectrogram. The first four separate types of BS comprise the single burst, multiple bursts, CRS, tone sound, harmonic sound, and the last one is their combination. Their behavior and characteristics were described, and the quantities of different types recorded from each participant were counted. These quantities varied significantly from person to person, from 0.04 to 0.7 per second. However, the proportions of different types of BS were reasonably stable. The mean proportion of SB sounds was 87.8% ± 5.4%, mean multiple bursts proportion was 7.8% ± 3.5%, mean continues random sound proportion was 3.5% ± 2.0%, and the mean proportion harmonic sound was 0.9% ± 0.5%. Also, from the eight- and four-hour recording, six MMC cycles were successfully observed based on the sound index with clear phases identifiable under the fasting state, and, as expected, the MMC disappeared once the participant consumed food. The MMC cycle was also observed from the two-hour-long recordings taken from ten other participants. This study also shows that when the gut enters MMC Phase II and III, longer BS were generated due to the increased gut activity by looking at the sound duration. This result shows that this simple and low-cost acoustic sensing device can effectively detect the MMC cycle by using the sound index as its biomarker. This work will help in further bowel sound studies with long recordings by providing guidance on the design of an effective acoustic sensing device and may aid in the development of new methods for understanding GI physiology.

## Figures and Tables

**Figure 1 sensors-18-04240-f001:**
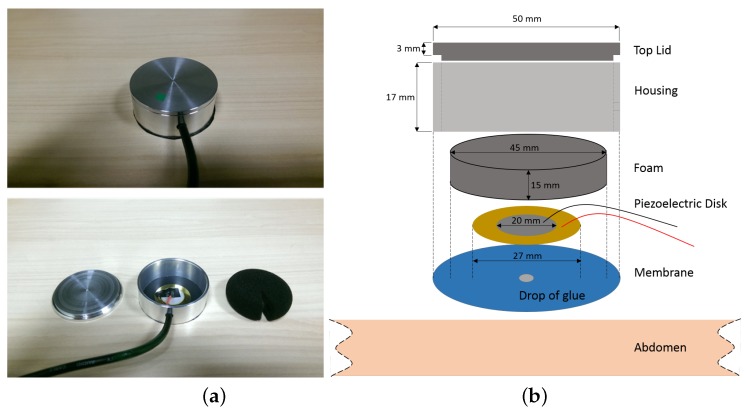
(**a**) Decomposition of the real sensor and (**b**) its corresponding consistence including the top lid, housing, foam, piezoelectric disk, and the membrane.

**Figure 2 sensors-18-04240-f002:**
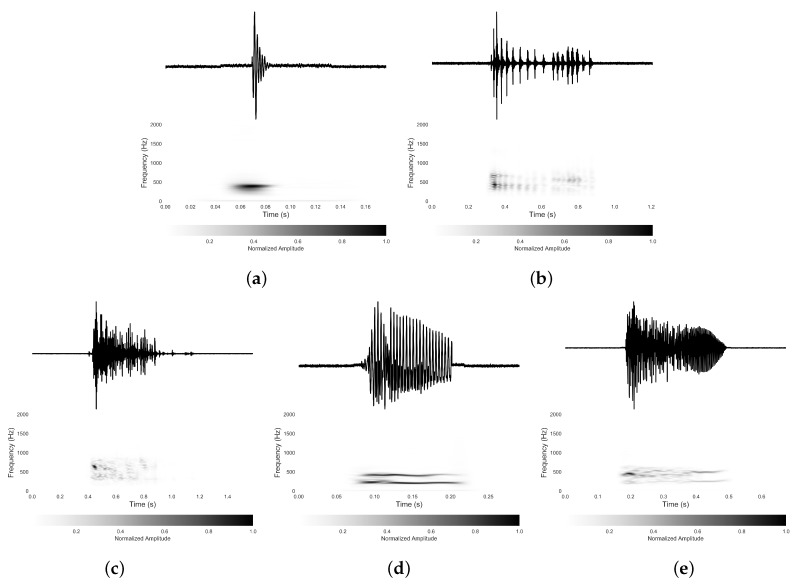
An example of the (**a**) single burst; (**b**) multiple bursts; (**c**) random continued sound; (**d**) harmonic sound; and (**e**) combination sound in time domain (**top**) and its corresponding spectrogram (**bottom**).

**Figure 3 sensors-18-04240-f003:**
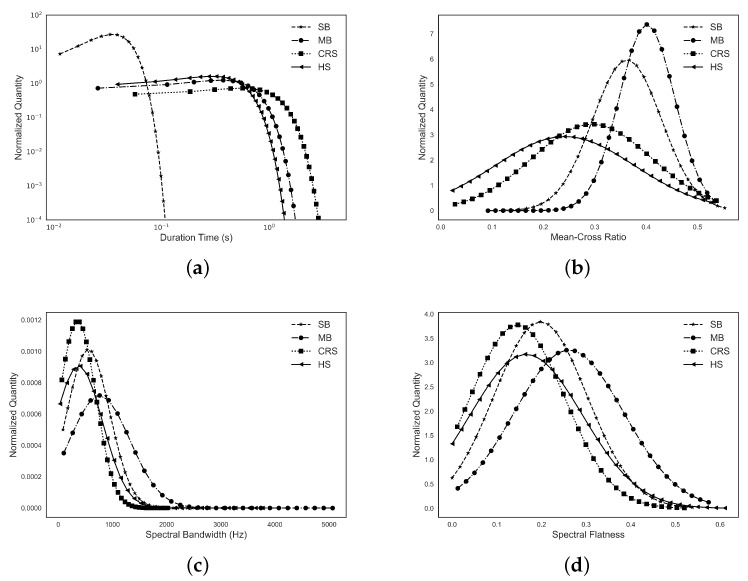
Normalized distribution (area under curve equals to one) of acoustics characteristics of four types of BS including (**a**) duration; (**b**) mean-cross ratio; (**c**) spectral bandwidth; and (**d**) spectral flatness.

**Figure 4 sensors-18-04240-f004:**
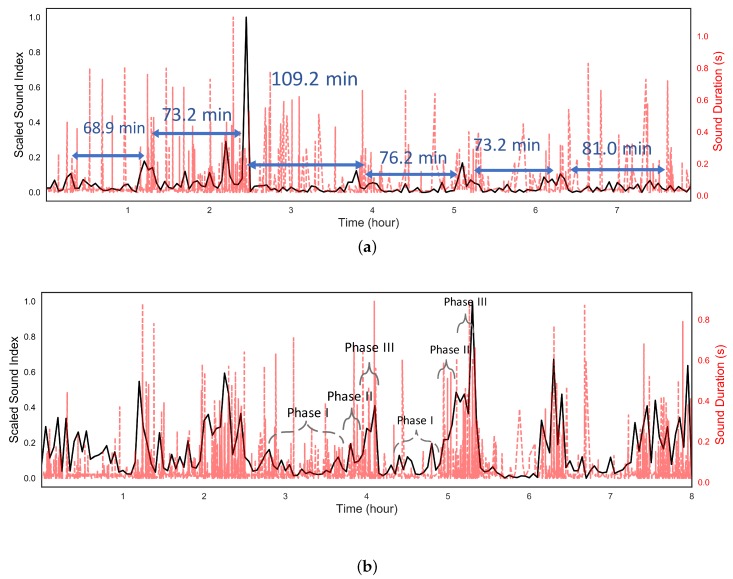
The scaled sound index of BS every three minutes (black curve) and sound duration (red curve) over eight hours under fasting at (**a**) upper quadrant and (**b**) lower quadrant.

**Figure 5 sensors-18-04240-f005:**
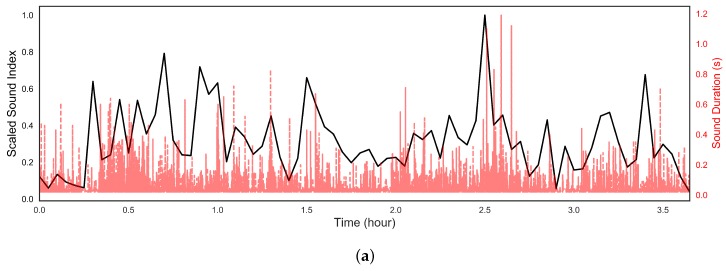
The scaled sound index of BS and the sound duration at lower quadrant (**a**) over four hours after meal from participant L and (**b**) over two hours under fasting stage from participant No. 1 to 3.

**Table 1 sensors-18-04240-t001:** The quantities and their corresponding proportion of different types of BS of 10 participants.

Participant No.	Gender	BMI	SB	MB	CRS	HS
1	female	36.7	~800 (86.4%)	82 (8.9%)	27(2.9%)	13 (1.3%)
2	male	38.1	~900 (89.9%)	65 (6.5%)	28 (2.8%)	6 (0.6%)
3	female	29.4	~3900 (96.8%)	74 (1.8%)	44 (1.1%)	4 (0.01%)
4	female	21.9	~2200 (84.1%)	354 (9.4%)	141 (5.4%)	11 (0.4%)
5	male	19.1	~1400 (76.7%)	245 (13.4%)	142 (7.8%)	23 (1.3%)
6	female	37.6	~2000 (88.0%)	151 (6.6%)	98 (4.3%)	13 (0.6%)
7	male	26.0	~3900 (89.5%)	208 (4.8%)	153 (3.5%)	77 (1.7%)
8	female	27.6	~250 (93.6%)	11 (4.1%)	4 (1.5%)	2 (0.7%)
9	female	22.8	~3900 (82.6%)	590 (12.5%)	149 (3.2%)	47 (1.0%)
10	female	28.4	~4500 (89.3%)	457 (9.1%)	50 (1.0%)	18 (0.4%)
Mean		28.8	2375.0 (87.8%)	223.7 (7.8%)	83.6 (3.5%)	21.4 (0.9%)
SD ^1^		6.4	1476.0 (5.4%)	180.1 (3.5%)	56.0(2.0%)	22.2 (0.5%)
CV ^2^		0.22	0.62 (0.06)	0.80 (0.45)	0.67 (0.57)	1.04 (0.56)

^1^ Standard Deviation; ^2^ Coefficient of Variation ($\frac{SD}{Mean}$).

**Table 2 sensors-18-04240-t002:** The acoustics features’ distributions of different types of BS.

Type	Duration (ms) 25–75%	Spectral Bandwidth (Hz) 25–75%	Spectral Flatness 25–75%	Mean-Crossing Ratio 25–75%
SB	26–42	283.7–710.9	0.1207–0.2585	0.3221–0.4059
MB	56–445.8	358.5–995.8	0.1647–0.3477	0.3681–0.4411
CRS	215–674	151.5–419.7	0.06818–0.2007	0.2096–0.3867
HS	124–385	145.6–438.1	0.07096–0.2272	0.1211–0.3493

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
