# Peer review of "Bowel Sounds Identification and Migrating Motor Complex Detection with Low-Cost Piezoelectric Acoustic Sensing Device"

_sensors, 2018, doi:10.3390/s18124240_

Round 1

Reviewer 1 Report

See attachment

Author Response

The point-by-point response to the reviewer's comment is uploaded as a PDF file

Reviewer 2 Report

In this paper, a straightforward piezoelectric acoustic sensor was designed and used for long-time bowel sounds (BS) recordings. Although piezoelectric acoustic sensors with similar configuration have been commonly applied in many occasions, the research in this paper is still quite meaningful, by which the piezoelectric acoustic sensor was expanded the application to the diagnosis of gastrointestinal (GI) conditions. In the paper, the migrating motor complex (MMC) cycle was detected using the piezoelectric sensor and the sound index as the biomarker for MMC phases. The measured acoustic characteristics and distribution are systematically described and analyzed, which verified that the sensors provide a useful tool for MMC detection and study of GI physiology and function.

Just like the authors’ comment “the exact structure of their sensor is never mentioned and their studies mainly focused on signal processing and BS types classification” on the previous researches, this paper is also mainly focused on signal and classification, though the structure, and only the structure of the sensor is mentioned. So I just suggest the authors to provide some detailed properties of the developed piezoelectric acoustic sensor, for example, the sensitivity and the ratio of signal/noise, which is closely related with the dimensions and the piezoelectric properties of the piezoelectric unit. These parameters are also important for the detection reliability and the flexibility to different persons.

Author Response

The point-by-point response to the reviewer's comments is uploaded as a PDF file

Round 2

Reviewer 1 Report

I am agree to publish the paper.